# Prevalence of SARS-CoV-2 in an area of unrestricted viral circulation: Mass seroepidemiological screening in Castiglione d'Adda, Italy

**Gabriele Pagani** [1,2]*, **Andrea Giacomelli** [1,2], **Federico Conti** [1,2], **Dario Bernacchia** [1,2], **Rossana Rondanin** [3], **Andrea Prina** [3], **Vittore Scolari** [4], **Arianna Rizzo** [1], **Martina Beltrami** [1], **Camilla Caimi** [1], **Cecilia Eugenia Gandolfi** [5,6], **Silvana Castaldi** [5,6], **Bruno Alessandro Rivieccio** [7], **Giacomo Buonanno** [8], **Giuseppe Marano** [9], **Cosimo Ottomano** [10], **Patrizia Boracchi** [9], **Elia Biganzoli** [9‡], **Massimo Galli** [1,2‡]

1 Dipartimento di Scienze Biomediche e Cliniche "L. Sacco", Università degli Studi di Milano, Milano, Italy, 2 Malattie Infettive III Divisione, ASST FBF-Sacco, Milano, Italy, 3 Medispa s.r.l., Milano (MI), Italy, 4 Institut Curie-PSL Research University, CNRS, Sorbonne Université, Paris, France, 5 Dipartimento di Scienze Biomediche per la Salute, Università degli Studi di Milano, Milano, Italy, 6 Fondazione IRCCS Ca' Granda Ospedale Maggiore Policlinico di Milano, Milano, Italy, 7 Dipartimento di Emergenza e Urgenza, ASST Grande Ospedale Metropolitano Niguarda, Milano, Italy, 8 Fondazione Istituto Insubrico di Ricerca per la Vita, LIUC Università Cattaneo, Castellanza, Italy, 9 Dipartimento di Scienze Cliniche e di Comunità, Laboratorio di Statistica Medica Biometria ed Epidemiologia "G.A. Maccacaro", Università degli Studi di Milano, Milano, Italy, 10 Synlab Italia s.p.a., Monza (MB), Italy

☺ These authors contributed equally to this work.
‡ Equally contributing Senior Authors.
* gabriele.pagani@unimi.it

**Data Availability Statement:** All relevant data are within the manuscript and its Supporting information files.

## Abstract

Castiglione D'Adda is one of the municipalities more precociously and severely affected by the Severe Acute Respiratory Syndrome Coronavirus 2 (SARS-CoV-2) epidemic in Lombardy. With our study we aimed to understand the diffusion of the infection by mass serological screening. We searched for SARS-CoV-2 IgGs in the entire population on a voluntary basis using lateral flow immunochromatographic tests (RICT) on capillary blood (rapid tests). We then performed chemioluminescent serological assays (CLIA) and naso-pharyngeal swabs (NPS) in a randomized representative sample and in each subject with a positive rapid test. Factors associated with RICT IgG positivity were assessed by uni- and multivariate logistic regression models. Out of the 4143 participants, 918 (22·2%) showed RICT IgG positivity. In multivariable analysis, IgG positivity increases with age, with a significant non-linear effect (p = 0·0404). We found 22 positive NPSs out of the 1330 performed. Albeit relevant, the IgG prevalence is lower than expected and suggests that a large part of the population remains susceptible to the infection. The observed differences in prevalence might reflect a different infection susceptibility by age group. A limited persistence of active infections could be found after several weeks after the epidemic peak in the area.

**Funding:** This study was realized thanks to non-conditioning financial contributions from: CISOM (Corpo Italiano di Soccorso dell'Ordine di Malta), FC Internazionale Milano, SFD s.p.a, Emporio Armani Olimpia Milano through donations to the Dipartimento di Scienze Biomediche e Cliniche (DIBIC) of the University of Milan; from Banca Mediolanum through donation to ASST Fatebenefratelli-Sacco, Milano. Mylan Italia s.p.a. donated the rapid tests. The donations were used to cover the expenses related to personal protective equipments, materials, laboratory processing and personnel costs. None of the funding sources were involved in data collection, analysis or intepretation; trial design; patient recruitment; or any aspect pertinent to the study. Medispa s.r.l. provided support in the form of salaries for authors RR and AP; Synlab s.p.a. provided support in the form of salaries for author CO; both funders did not have any additional role in the study design, data collection and analysis, decision to publish, or preparation of the manuscript. The specific roles of these authors are articulated in the 'author contributions' section.

**Competing interests:** Medispa s.r.l. provided support in the form of salaries for authors RR and AP; Synlab s.p.a. provided support in the form of salaries for author CO. In particular none of the commercial funders (FC Internazionale Milano, Emporio Armani Olimpia Milano, SFD s.p.a. and Banca Mediolanum s.p.a.) had any role in data collection, analysis or intepretation; trial design; patient recruitment; or any aspect pertinent to the study, nor they have any commercial interest (e.g. consultancy, patents, products in development, marketed products, etc). This does not alter our adherence to PLOS ONE policies on sharing data and materials.

## Introduction

Italy is the European country in which the Severe Acute Respiratory Syndrome Coronavirus 2 (SARS-CoV-2) epidemic had the earliest expansion, probably starting from the last days of January [1]. From the end of February to mid-June 2020, more than 240,000 confirmed cases and more than 34,000 deaths have been reported [2].

The true number of SARS-CoV-2 infections is estimated to be several times higher than the official one, mainly due to molecular testing being restricted to hospitalized and severely symptomatic cases. The available seroprevalence studies on SARS-CoV-2 enrolled special populations, such as healthy blood donors [3], healthcare workers [4] or hospitalized patients [5, 6], and consequently are not easily generalizable to the entire population. Interestingly, recent nation-wide studies have widely different estimated seroprevalences, ranging from 1% to 6·9% in the United States [7] and 5% in Spain [8].

The WHO considers forwarding scientific knowledge on mass screening essential for the battle against the spread of SARS-CoV-2, in order to allow better understanding and planning of current and future containment policies [9]. Recently, lateral flow immunochromatographic tests on capillary blood (rapid immunochromatographic tests, RICT) have been proposed as point-of-care serological assays. They have been recently used in a large seroepidemiological study in Spain, demonstrating high accuracy, while having a greater uptake, lower cost, and easier implementation compared to other diagnostic methods [8].

Castiglione d'Adda (CdA) is a town of 4605 inhabitants (according to data from the local registry office at the time of our study), situated about 30 km South-East of Milan on the banks of the Adda river. Local economy is mainly based on farming and small industries. The part of the population working in other fields mainly commutes to Milan, Lodi or other larger neighboring cities. The municipality of CdA has been heavily affected by SARS-CoV-2 infection since the earliest stages of the epidemic: the first Italian patient hospitalized for Coronavirus Disease-19 (COVID-19) was a citizen of CdA. Since February 23rd, 2020 the town was included among the first so-called "red zone" and its population subjected to lockdown [10].

Of the 3412 confirmed COVID-19 cases reported in the province of Lodi [11, 12], 184 were diagnosed in people living in CdA, accounting for around 4% of the resident population. From the 1st of January to the 31st of March 2020, 76 deaths (1·65% of the population) have been recorded in CdA, of which 47 were officially attributed to COVID-19.

The aim of our study was to estimate the seroprevalence of SARS-CoV-2 infection and the epidemiological characteristics of the infected population in an epidemic setting characterized by initial unrestricted circulation of SARS-CoV-2. An integrated approach based on RICTs, chemiluminescent immunoassays (CLIA) serologies and real time reverse transcriptase polymerase chain reaction (RT-PCR) on naso-pharyngeal swabs (NPS) was applied.

## Material and methods

### Objectives

The primary objective of the study was to assess the seroprevalence for SARS-CoV-2 infection in CdA. The secondary objectives were to characterize the self-reported symptoms in those with and without a positive serology; to search for factors associated to SARS-CoV-2 IgG seropositivity; to assess the hospitalization rate of seropositive subjects and to estimate the infection fatality rate; and to assess the diagnostic performance of RICT when compared to CLIA

## Study design

A cross-sectional study was conducted from the 18th of May to the 7th of June. The entire population of CdA was offered to participate by dissemination of the news using the municipality website and informative sheets in public locations.

A random representative sample, stratified by age and gender, underwent venipuncture for CLIAs and NPS for RT-PCR regardless of RICT result.

## Study procedures

The study was based on an integrated approach including screening by RICTs and subsequent confirmation of positive cases by CLIA serologies. RT-PCR on NPSs was performed in all the RICT-positive subjects to exclude ongoing viral shedding. Study participants were invited to be tested (not more than 10 people per half-hour shifts, to ensure social distancing, approximately 200–250 people per day) through a booking system managed by the municipal administration, 12 hours per day and 7 days per week. A standard questionnaire containing epidemiological, clinical and anamnestic information was administered prior to testing. The full list of questions is provided in the S1 Appendix. RICTs were read by experienced health personnel. Venous blood drawing and NPSs were performed by skilled nurses. A medical doctor was present on location for the whole duration of the study (both for emergencies and counselling purposes). Children under 12 years of age were tested in "pediatric shifts", when a pediatric nurse and a pediatrician were present.

Positive NPSs were reported through the Regional surveillance system, quarantined and tested again after 14 days [13]. People with documented past infection (at least a positive NPS) who had ended their quarantine and already reported two negative NPSs did not undergo RT-PCR in our study but underwent CLIA serologies.

## Samples' collection and handling

Samples (both NPSs and venous blood) were stored in a +4˚C refrigerator which was present on location and collected daily (except for Sunday and festivities) by Synlab's specialized courier. PCR and serologies were performed the day after collection and results were usually available the next working day from collection. Samples were identified by a label containing a unique barcode (as well as name, surname and date of birth of the subject) generated by Synlab through their proprietary software. Results were nominal and available to the attending physician only, who then communicated them to patients and, in case of a positive PCR, to their general practitioner.

## Detection of SARS-CoV-2 directed antibodies and viral RNA

Prima Lab SA Covid-19 IgG/IgM Rapid test (Prima Lab, Switzerland) was used as RICT. According to a recent review on the diagnostic performance of serological tests, these tests showed a sensitivity of 100% in detecting IgG antibodies more than 14 days after the infection and a specificity of 96%. IgM accuracy was lower, with less than 60% sensitivity and 93% specificity [14].

To confirm RICT results we used serological chemiluminescent microparticle immunoassay (CLIA) for qualitative detection of IgG against SARS-CoV-2 nucleoprotein on venous blood (SARS-CoV-2 IgG for use with ARCHITECT; Abbott Laboratories, Abbott Park, IL, USA). The manufacturer reported a sensitivity of 86·4% after 7 days from symptom onset and 100% after 14 days, and a specificity of 99·6%, using RT-PCR as the gold standard. IgG results on RICT were interpreted as "positive" (a clearly visible IgG band together with the control

band), "negative" (no IgG band with a visible control band) and "unclear" (a barely visible IgG band together with a control band). RICTs not showing a control band were discarded and the test was repeated using a new kit. "Unclear" results underwent CLIAs and RT-PCR for SARS-CoV-2 but were considered as negative for the purpose of our study.

Only IgG positivity was accounted as a "positive" result for analysis.

Finally, we used either TaqPath COVID-19 CE-IVD (ThermoFisher Scientific, USA) or RADI COVID-19 (KH Medical Co., Republic of Korea) RT-PCR detection kits to process NPSs, depending on local availability of reagents during the first wave of pandemic. In particular, the latter was used from the start of the study to the 25th of May, and then substituted with the former one. Accuracy was comparable [15, 16].

Any subject who reported one or more signs/symptoms including fever, cough, anosmia, dysgeusia, dyspnea, new-onset acute arthromyalgia or rash (in a period ranging from the 1st of February 2020 to the end of the study) was considered as "symptomatic".

Body Mass Index was calculated for subjects over 19 years of age (as per WHO's indications) and defined as a person's weight in kilograms divided by the square of the person's height in meters ($kg/m^2$) [17].

## Statistical analysis

Numerical variables were summarized using mean and standard deviation (after checking the symmetry of the respective distributions); categorical variables were summarized by total counts and percentages.

Estimated seroprevalence, defined as the proportion of subjects positive to IgG antibodies, was calculated with respective 95% CIs, using the binomial distribution. Further analyses were aimed at evaluating the association of RICT IgG positivity with the following factors: gender, age, contact with a confirmed covid-19 case, being a current smoker, being affected by chronic lung diseases, hypertension, other cardiovascular diseases, rheumatic diseases, diabetes mellitus, oncological pathologies, and presence of the following symptoms: fever, cough, anosmia, dysgeusia, dyspnea, rash, arthromyalgia, other symptoms. The association was evaluated by logistic regression models, with IgG positivity as response variable and the abovementioned factors as independent variables. Concerning age, a non-linear relationship with positivity to IgG was assessed by including restricted cubic splines with three knots in the respective model, and by testing the contribution of the non-linear term of the spline by the Wald test. Results were reported in terms of estimated unadjusted Odds Ratios (ORs) and estimated Adjusted Odds Ratios (aORs), with respective 95% CIs. To account for the joint contribution of the independent variables, the AORs were calculated by a multivariate regression approach. In a first step, a multivariate model including all the variables was fitted. To highlight the factors with significant multivariate association with positivity to IgG, a backward model selection procedure was used. Age and gender kept in the model regardless of univariate analysis, thus they were not subjected to the model selection procedure. For the remainder factors, the estimates of AORs were reported only for those that were kept within the model by the selection procedure.

The diagnostic accuracy of the RICT for IgG antibodies was evaluated by using IgG serological test results as reference method ("gold standard"). To such end, data of 509 subjects within the random representative sample (see: Study Design) with available results of both rapid and serologic tests were used. Estimates of sensitivity, specificity, with respective 95% CIs were obtained by logistic regression models estimated by Generalized Estimating Equation method [18] to account for the stratified sampling design.

For further analysis, infection fatality rate (IFR) was defined as the proportion of deceases over the total number of IgG positive subjects (rapid test) and estimated using the binomial distribution.

Statistical analysis was performed using the software R [19] v3·6·2 and Knime Analytics Platform-3·6·0 [20].

### Ethics statement

The study was approved by University of Milan's Ethical Committee (allegato 1 Comitato Etico 21.04.20—parere numero 35/20). A written informed consent, approved by the ethical committee, was signed by every subject participating in the study (or their parents or legal representants in case of minors of age). Data were fully anonymized before analysis.

## Results

A total of 4143 persons voluntarily participated to the study. Three-thousand-seven-hundred-and ninety-seven (3797, 91·6% of total participants) were residents, including the 39 hosts of the local residential care facility, accounting for 82·4% of the official resident population. The remaining 346 subjects were either non-resident inhabitants, domiciled within the municipality (n = 43) or people working in CdA on a daily basis during the epidemic period (for example healthcare workers, police and military officials, etc.; n = 303).

### Demographics

Demographic and clinical characteristics of the screened population are reported in Table 1. Mean age was 48 years (Standard Deviation [SD]: 20·9 years; range: 0–102 years) and females represented 51·3% (n = 2125) of the tested population. IgG positive subjects were more frequently non-smokers compared to IgG negative ones (21·2% *vs* 12·3%, p<0·0001).

Sixty-nine (1·6%) participants reported hospital admission due to SARS-CoV-2 infection.

The proportion of asymptomatic infections (people with an IgG positive RICT who didn't report any "typical" symptom [fever, cough, dyspnea, anosmia, dysgeusia, arthromyalgias and rash]) was 30·5% (280/918). Conversely, people who did not report any symptom were more frequently found among RICT IgG negative participants (72·9% vs. 30·5%, p<0·0001).

Demographic and clinical characteristics of the random sample, used to estimate prevalence based on CLIAs, are reported in S1 Table.

### Seroprevalence and active infections

Seroprevalence based on CLIA serologies of the 509-subjects random sample was 22·6% (95% CI 17·2%, 29·1%).

In the overall population 918 (22·2%) RICT showed a positivity on the IgG band, while 23 (0·6%) were interpreted as unclear.

We found 22 positive RT-PCR in 1330 NPSs performed; this number includes both NPS performed in RICT-positive subjects and in randomized subjects. It is worth noting that the two groups overlap, since a randomized subject could be RICT positive due to *a priori* randomization. More specifically, among the subjects with a positive NPS, 20 had a positive RICT, while 2 subjects had a negative RICT and were selected for the random sample.

### Factors associated with IgG positivity

The estimates of unadjusted and adjusted Odds ratios from univariate and multivariate regression analysis are reported in Table 2.

**Table 1. Characteristics of 4143 study participants.**

| | IgG negative/unclear (n = 3225) | IgG positive (n = 918) | Total Sample (n = 4143) |
|---|---|---|---|
| **Sex (Male)** | 1600 (49·6%) | 418 (45·5%) | 2018 (48·7%) |
| **Age (years)** | 46·0, 20·8 | 55·2, 20·0 | 48·0, 20·9 |
| **Residence status:** | | | |
| Residents | 2937 (91·1%) | 821 (89·5%) | 3758 (90·7%) |
| RCF | 0 (0·0%) | 39 (4·2%) | 39 (0·9%) |
| Non residents | 288 (8·9%) | 58 (6·3%) | 346 (8·4%) |
| **Contact with verified case** | 723 (22·4%) | 467 (50·9%) | 1190 (28·7%) |
| **BMI (Kg/m$^2$)** | 24·6, 4·3 | 24·9, 4·9 | 24·7, 4·4 |
| **Smoker** | 685 (21·2%) | 113 (12·3%) | 798 (19·3%) |
| **Cardiovascular diseases:** | | | |
| • CAD/MI | 71 (2·2%) | 23 (2·5%) | 94 (2·3%) |
| • Arrhythmias | 105 (3·3%) | 48 (5·2%) | 153 (3·7%) |
| • Hypertension | 562 (17·4%) | 287 (31·3%) | 849 (20·5%) |
| • Other | 180 (5·6%) | 81 (8·8%) | 261 (6·3%) |
| At least one of the above: | 744 (23·1%) | 350 (38·1%) | 1094 (26·4%) |
| Rheumatic diseases | 159 (4·9%) | 62 (6·8%) | 221 (5·3%) |
| Diabetes mellitus | 129 (4·0%) | 57 (6·2%) | 186 (4·5%) |
| **Chronic Lung diseases** | | | |
| • Asthma | 115 (3·6%) | 28 (3·1%) | 143 (3·5%) |
| • COPD | 21 (0·7%) | 23 (2·5%) | 44 (1·1%) |
| • Other | 71 (2·2%) | 25 (2·7%) | 96 (2·3%) |
| At least one of the above: | 200 (6·2%) | 72 (7·8%) | 272 (6·6%) |
| **Oncological pathologies:** | | | |
| Solid tumors | 129 (4·0%) | 60 (6·5%) | 189 (4·6%) |
| Oncochematological | 28 (0·9%) | 6 (0·7%) | 34 (0·8%) |
| At least one of the above: | 155 (4·8%) | 65 (7·1%) | 220 (5·3%) |
| **Symptoms:** | | | |
| • Fever | 500 (15·5%) | 491 (53·5%) | 991 (23·9%) |
| • Cough | 430 (13·3%) | 259 (28·2%) | 689 (16·6%) |
| • Anosmia | 93 (2·9%) | 267 (29·1%) | 360 (8·7%) |
| • Dysgeusia | 103 (3·2%) | 311 (33·9%) | 414 (10·0%) |
| • Dyspnea | 128 (4·0%) | 135 (14·7%) | 263 (6·3%) |
| • Rash: | 54 (1·7%) | 46 (5·0%) | 100 (2·4%) |
| • Arthromyalgias | 242 (7·5%) | 258 (28·1%) | 500 (12·1%) |
| At least one of the above: | 878 (27·1%) | 638 (69·5%) | 1516 (36·6%) |
| Other symptoms | 290 (9·0%) | 227 (24·7%) | 517 (12·5%) |

Numerical variables (namely: age and BMI) are presented as means and standard deviations. Categorical variables are presented as total counts and percentages. BMI was calculated for 3668 subjects aged ≥20 years. RCF: Residential Care Facilities; BMI: Body Mass Index; CAD: Coronary Artery Disease; MI: Myocardial Infarction; COPD: Chronic Obstructive Pulmonary Disease.

In multivariate analysis, IgG positivity increases with age, with a significant non-linear effect included in the model (p = 0·0404). To show this relationship, estimates of aORs were calculated from the model selected by the backward procedure, by taking 65 years of age as reference. The estimates are shown in Fig 1. For example, for a subject aged 30 years the aOR is equal to 0·39 (95% CI: 0·33–0·47), and for subjects aged 50 and 70 years the aOR is equal to 0·62 (95% CI: 0·55–0·69) and 1·20 (95% CI: 1·14–1·26), respectively.

**Table 2. Association between positivity to IgG for RICT and the characteristics of interest.**

| | Unadjusted OR | | Adjusted OR | | | |
| --- | --- | --- | --- | --- | --- | --- |
| | | | Initial model | | Final model | |
| | Est (95% C.I.) | p-value | Est (95% C.I.) | p-value | Est (95% C.I.) | p-value |
| **Sex**: Female vs. male | 1·18 (1·02, 1·36) | 0·0291* | 0·90 (0·75, 1·08) | 0·2508 | 0·90 (0·75, 1·08) | 0·2466 |
| **Age** | | | | | | |
| Linear term | 1·25 (1·20, 1·30) | <0·0001* | 1·18 (1·06, 1·31) | 0·0037* | 1·18 (1·06, 1·32) | 0·0022* |
| Non-linear term | | | 1·15 (1·02, 1·29) | 0·0184* | 1·12 (1·00, 1·26) | 0·0404* |
| **Status**: Resident | 0·83 (0·65, 1·06) | 0·1373 | 1·03 (0·76, 1·41) | 0·8701* | | |
| **Contact with verified case** | 3·58 (3·07, 4·18) | <0·0001* | 2·84 (2·35, 3·42) | <0·0001* | 2·85 (2·37, 3·43) | <0·0001* |
| **Smoker** | 0·52 (0·42, 0·64) | <0·0001* | 0·67 (0·52, 0·86) | 0·0018* | 0·66 (0·51, 0·85) | 0·0016* |
| **Cardiovascular diseases** | | | | | | |
| • Hypertension | 2·16 (1·82, 2·54) | <0·0001* | 1·35 (1·06, 1·71) | 0·0139* | 1·32 (1·04, 1·67) | 0·0202* |
| • CAD/arr/MI/Other | 1·63 (1·31, 2·01) | <0·0001 | 0·95 (0·72, 1·25) | 0·7051 | | |
| **Rheumatic diseases** | 1·40 (1·02, 1·88) | 0·0347* | 0·83 (0·57, 1·20) | 0·3412 | | |
| **Diabetes mellitus** | 1·59 (1·15, 2·18) | 0·0060* | 0·94 (0·63, 1·39) | 0·7865 | | |
| **Chronic Lung diseases** | | | | | | |
| At least one | 1·29 (0·97, 1·69) | 0·0824 | 0·82 (0·57, 1·17) | 0·2876 | | |
| **Oncological pathologies** | | | | | | |
| At least one | 1·51 (1·11, 2·03) | 0·0087* | 0·75 (0·50, 1·09) | 0·1391 | | |
| **Fever** | 6·27 (5·34, 7·37) | <0·0001* | 4·10 (3·29, 5·11) | <0·0001* | 4·08 (3·34, 4·97) | <0·0001* |
| **Cough** | 2·55 (2·14, 3·04) | <0·0001* | 0·78 (0·60, 1·00) | 0·0564 | | |
| **Anosmia** | 13·81 (10·79, 17·82) | <0·0001* | 1·81 (1·12, 2·89) | 0·0140* | 1·84 (1·15, 2·93) | 0·0108* |
| **Dysgeusia** | 15·53 (12·27, 19·80) | <0·0001* | 5·57 (3·60, 8·73) | <0·0001* | 5·66 (3·67, 8·81) | <0·0001* |
| **Dispnea** | 4·17 (3·24, 5·38) | <0·0001* | 1·21 (0·85, 1·72) | 0·2877 | | |
| **Rash** | 3·10 (2·07, 4·62) | <0·0001* | 1·65 (0·95, 2·83) | 0·0705 | | |
| **Arthromyalgia** | 4·82 (3·97, 5·86) | <0·0001* | 1·25 (0·95, 1·63) | 0·1084 | | |
| **Other symptoms** | 3·32 (2·74, 4·03) | <0·0001* | 1·30 (1·01, 1·68) | 0·0438* | 1·34 (1·04, 1·72) | 0·0224* |

Initial model: multivariate model including all the considered factors included as independent variables. Final model: Model selected by the backward selection procedure. Linear term and non-linear term refer to the regression model coefficients required for the estimation of the non-linear effect of age. OR: Odds Ratio, Est: estimate, C.I.: Confidence interval. RCF: Residential Care Facilities; BMI: Body Mass Index; CAD: Coronary Artery Disease; MI: Miocardial Infarction; COPD: Chronic Obstructive Pulmonary Disease.

As shown in Table 2, the backward selection procedure confirmed factors associated with IgG positivity in the initial multivariate model.

In particular, having had a contact with a verified case was associated with a higher probability of having a positive RICT (aOR 2·85, 95%CI 2·13–2·43, p<0·0001), while being an active or former smoker was associated with a lower probability of having a positive RICT (aOR 0·66, 95%CI 0·51–0·85, p = 0·0016) with respect to never-smokers.

Of note is the observation that a significant association with IgG positivity was found for every symptom in univariate analysis, with a higher strength of association with anosmia and dysgeusia (anosmia: OR = 13.8; 95% CI 12.3 to 19.8; dysgeusia: OR = 15.5; 95% CI: 12.3,19.8). In multivariate analysis, a significant association was found for fever, anosmia, dysgeusia and other symptoms, but not for the remaining ones. Regarding the morbidities, in univariate analysis a significant association was found for each morbidity group considered, except chronic lung diseases, whereas in multivariate analysis only hypertension showed a significant association with IgG positivity (aOR 1·32, 95% CI: 1·04, 1·67, p = 0·02).

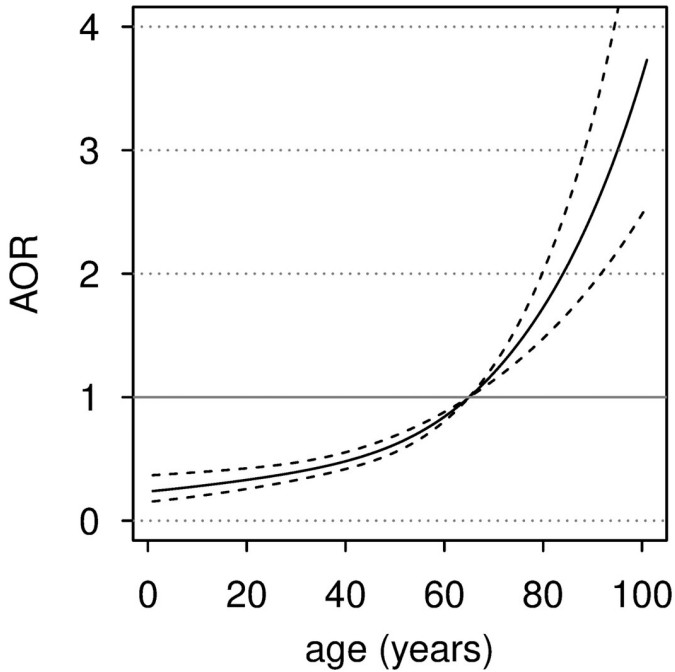

**Fig 1.**

Similar results were obtained fitting the model on the random representative sample (S2 Table); although, in case of age, the non-linear relationship with IgG positivity on CLIAs was not significant.

### Infection fatality rate

Based on our seroprevalence estimate by RICT and the 47 deaths officially recorded as COVID-19-related, we found an infection fatality rate of 5·1% (95% CI: 3·9%, 6·7%).

### Diagnostic accuracy

Five hundred and sixty-two (562) subjects were randomly selected for receiving CLIA serological tests and PCR on NPS, regardless of their RICT results. CLIA serologies were performed in 509. Fifty-three (53, 9·4%) were excluded from analysis either because they refused to undergo venipuncture or because of pre-analytical or analytical errors in sample processing. Based on this subset of patients we estimated RICT diagnostic accuracy, considering CLIA serologies as the gold-standard: relative specificity was 95·9% (95% CI: 94·8%, 96·8%) and relative sensitivity was 97·4% (95%CI: 90·9%, 99·3%).

### Discussion

To the best of our knowledge, this is the first SARS-CoV-2 seroepidemiological study set in a zone of unrestricted viral circulation. Approximately 22% of CdA resident resulted to have been exposed to SARS-CoV-2 infection. On the contrary, the number of positive NPSs was very low, suggesting a strongly reduced viral circulation at the time of our study. Thus, it could be speculated that the persistency of active infections, which could not be detected by antibody testing, only marginally influenced our estimates on the actual spread of SARS-CoV-2.

Due to the limited performance of RT-PCR on NPS as a tool in mass population screening [21, 22], serological tests on venous blood (such as Enzyme Linked Immuno Assays [ELISA] [7] or CLIA [23]) have been used for this purpose. Nevertheless, these methods also present important limitations: blood drawing needs trained personnel in a dedicated location, it's time consuming for both operators and tested persons and serology processing requires specialized laboratory equipment with long turnaround time. Moreover, a non-negligible portion of candidates of a mass population screening is reluctant to accept venipuncture. On the contrary RICTs offer a rapid, minimally invasive, point-of-care alternative, suitable for screening a large amount of people in a short time with good diagnostic accuracy, especially for IgG [8]. In our study, the adopted RICT showed a specificity of 95·9% and sensitivity of 97·4% when compared to CLIA. According to the screening strategy reported by Pollán and colleagues in the ENE-COVID study [8], we decided to consider only IgG positivity, both on RICTs and CLIAs. The prevalence based on RICTs (22·2%) is similar to that obtained by CLIA serologies on the random stratified sample (22·6%) and confirms the estimate obtained by Percivalle *et al*. in their study on healthy blood donors set in the same geographical area [3]. This is a lower-than-expected prevalence, considering that it was recorded in one of the most severely-affected areas in Italy. It is worth noting, however, that it is based on antibody detection only, while recent evidence suggests that a robust T-cell-mediated immunity could be present also in sero-negative individuals [24].

The strong association observed between IgG seroprevalence and age deserves some consideration: the fact that 10 year-old children would have a 0·28 aOR to be seropositive, while 95-year-old subjects would have a 2·5-fold association to IgG positivity, when compared to 65-year old, strongly suggest a different age-related susceptibility to the infection.

Differences in seroprevalence related to age were also noticed in a large seroprevalence study set in Spain [8] and in the study by Havers *et al*. conducted in the USA [7]. In both studies, the lowest prevalence was found among the youngest individuals, even if the eldest subjects had a lower prevalence than the middle-aged ones. However, it should be remembered that the virus was able to circulate in CdA for several weeks before containment measures were introduced, while both Spain and the US were hit by the pandemic when it was already clear that the infection was particularly dangerous for the elderly. Therefore, it cannot be excluded that spontaneous precautionary measures have reduced the spread of infection in this age group. Qualitative and quantitative difference in ACE2 receptor [25] could be considered as a hypothetical cause of age-related differences in seroprevalence. Younger subjects supposedly have a minor expression of ACE2 [26], thus resulting less susceptible. Since the assumption of an age-related increased susceptibility to SARS-CoV-2 infection could be true if different age groups had the same probability to be exposure to the infection, it must be acknowledged that, in Italy, schools of any grade were closed a week before the general lock-down. Nevertheless, CdA was exposed to unrestricted viral circulation for weeks before the lock-down measures were enforced, which probably renders the effect of early school closure less relevant.

Surprisingly, reported smoking (either current or former) was associated with a lower probability of being IgG seropositive. However, a possible protective effect of cigarette smoking against infection has already been postulated. In particular, in the pooled data of 12 studies, current smokers showed a 0·70 RR of becoming infected (95%CI 0·55–0·88) [27]. A nicotinergic downregulation of ACE2 receptors in lower airways has been hypothesized [28], but further studies are needed to explain such a phenomenon.

Hypertension was also weakly, but significantly and independently associated to having a positive RICT (aOR 1·32). Renin–Angiotensin–Aldosterone System Inhibitors, commonly used to treat hypertension, have a role in ACE2 homeostasis and it is possible that a modification in quantity and site of ACE2 expression could modify susceptibility and disease outcome

[29]. However, our questionnaire did not focus on home therapies and consequently this hypothesis cannot be confirmed with our study.

Regarding symptoms, the association between olfactory and taste disorders and seropositivity was an expected finding: preliminary observations of an association of self-reported olfactory and taste disorders with SARS-CoV-2 infection in hospital settings [30] have now been confirmed in the general Italian population, as observed in a large web-based nation-wide survey [31]. Even with the limitations of a study based on participants' reports weeks after the peak of the epidemic, in which a recall bias could occur, a high percentage of positive cases did not report any symptoms related to COVID. On the other hand, one must remember that in CdA the psychological impact of COVID was so strong as to probably lead the population to consider and remember every symptom attributable to the disease, strongly reducing the probability of recall bias. Thus, the 30% of IgG positive subjects who did not report any symptom in the previous months could represent an accurate estimate of the percentage of asymptomatic infections in an area with unrestricted viral circulation. Interestingly, a nearly identical percentage of asymptomatic infections was observed in the Spanish study [8].

The precise estimation of the real infection lethality remains a matter of debate [32]. During the early wave of the Italian epidemic some estimates of the case fatality rate (7·2%) [33] and intra-hospital 30-day mortality (20·6%) [34] have been provided. The absence of systematic testing and contact tracing strategies, however, did not allow to provide valid estimates of the true infection fatality rate, especially outside the healthcare setting. The 5% infection fatality rate (IFR) estimated in our study is the first reported in Italy. IFR estimates in other countries, however, are markedly lower and range from 0·36% in a small German town [35], to 0·58% in Indiana [36], and 0·66% in China [37]. While our figure should be considered with caution, awaiting other estimates from areas less affected by the epidemic, it cannot be excluded that the demographic characteristics of the Italian population, with a high proportion of elderly people, could have led to an increased IFR. Also, we cannot rule-out that the dramatic situation of the first weeks of the epidemic could have delayed hospitalization in some cases, with negative effects on the probability of survival. These differences, however, could be also influenced by different national death-reporting systems. It is worth noting that all reported estimates so far are higher than those reported for influenza (IFR: 0·1%) [38].

## Limitations and strengths

Our study presents several limitations.

Signs, symptoms and epidemiological and anamnestic characteristics (i.e. chronic diseases) were self-reported through a questionnaire and may not be completely accurate. Additionally, although it is not possible to exclude "voluntary bias" (e.g. previously symptomatic people were more likely to participate, and had a higher probability to test positive), the almost complete participation of the CdA population in our screening should make it less relevant. Moreover, while being the only tests readily available for mass screening as of today, serologies (both CLIAs and RICTs) may not be the optimal means to detect past infections because of variable humoral response. In particular, recent literature suggests that SARS-CoV-2 antibody levels could wane over time, especially in asymptomatic individuals [39]. Our study, however, started roughly three months after the first recorded case and two months after the start of the national lockdown, thus we do not expect the humoral response to be significantly diminished in the general population.

Finally, the study was conducted in the area of first epidemic expansion, characterized by an initially uncontrolled viral circulation. While this is a limit to the generalizability of the study, it offers a unique opportunity to assess the detrimental impact of unrestricted viral circulation in a fully susceptible population.

## Conclusions

In conclusion, we found that less than 23% of the CdA population has detectable SARS-CoV-2-directed IgG antibodies, thus leaving most of the population susceptible to infection despite being one of the most severely affected areas in Italy and, probably, the world. Seroprevalence significantly increases with age, suggesting a lower susceptibility to infection in infants and children. The high estimated infection fatality rate (5%), paired with the low prevalence of SARS-CoV-2 antibodies, warrants the maintenance of protective and distancing measures for the frailer part of the population and an immediate reinforcement of diagnostic and surveillance capacities at a territorial level.

## Supporting information

**S1 Table. Characteristics of 509 subjects in the random sample.** Numerical variables (namely: age and BMI) are presented as means and standard deviations. Categorical variables are presented as total counts and percentages. BMI was calculated for 3668 subjects aged $\geq$20 years. RCF: Residential Care Facilities; BMI: Body Mass Index; CAD: Coronary Artery Disease; MI: Miocardial Infarction; COPD: Chronic Obstructive Pulmonary Disease.
(DOCX)

**S2 Table. Association between positivity to IgG for CLIAs and the characteristics of interest.** Initial model: multivariate model including all the considered factors included as independent variables. Final model: Model selected by the backward selection procedure. Linear term and non-linear term refer to the regression model coefficients required for the estimation of the non-linear effect of age. OR: Odd Ratio, Est: estimate, C.I.: Confidence interval. BMI: Body Mass Index; CAD: Coronary Artery Disease; MI: Myocardial Infarction; COPD: Chronic Obstructive Pulmonary Disease.
(DOCX)

**S1 Appendix. Administered epidemiological and clinical questionnaire.**
(DOCX)

**S1 Data.**
(XLS)

## Acknowledgments

The authors would like to thank all healthcare and non-healthcare personnel who worked hard, often on a voluntary basis, to make this study possible. In particular, we would like to thank all volunteer nurses and law enforcement, Costantino Pesatori, the Major of CdA, and all the municipal administration for their enthusiastic partecipation and precious help.

We would also like to thank Dr. Alberto Saracco, MD, for his suggestions during the definition of the protocol, and his tireless help during the practical organization of the study; and Bianca Ghisi and Tiziana Formenti, for their valuable technical help. Finally, we thank dr. Alessandro Visconti, General Manager of ASST Fatebenefratelli-Sacco, Milano, and Dr. Roberto Infurna and Stefania Vimercati for the precious collaboration.

## Author Contributions

**Conceptualization:** Gabriele Pagani, Andrea Giacomelli, Elia Biganzoli, Massimo Galli.

**Data curation:** Rossana Rondanin, Andrea Prina, Vittore Scolari, Arianna Rizzo, Martina Beltrami, Camilla Caimi, Bruno Alessandro Rivieccio, Giacomo Buonanno.

**Formal analysis:** Giacomo Buonanno, Giuseppe Marano, Patrizia Boracchi, Elia Biganzoli.

**Funding acquisition:** Massimo Galli.

**Investigation:** Federico Conti, Vittore Scolari, Arianna Rizzo, Martina Beltrami, Camilla Caimi.

**Methodology:** Gabriele Pagani, Andrea Giacomelli, Federico Conti, Dario Bernacchia, Giuseppe Marano, Elia Biganzoli.

**Project administration:** Gabriele Pagani, Federico Conti, Rossana Rondanin, Massimo Galli.

**Resources:** Rossana Rondanin, Andrea Prina, Cosimo Ottomano.

**Software:** Rossana Rondanin, Andrea Prina.

**Supervision:** Gabriele Pagani, Andrea Giacomelli, Dario Bernacchia, Silvana Castaldi, Elia Biganzoli, Massimo Galli.

**Validation:** Andrea Giacomelli, Vittore Scolari, Cecilia Eugenia Gandolfi, Silvana Castaldi, Giacomo Buonanno, Patrizia Boracchi, Elia Biganzoli, Massimo Galli.

**Visualization:** Giuseppe Marano.

**Writing – original draft:** Gabriele Pagani, Andrea Giacomelli, Massimo Galli.

**Writing – review & editing:** Gabriele Pagani, Andrea Giacomelli, Federico Conti, Dario Bernacchia, Cecilia Eugenia Gandolfi, Silvana Castaldi, Giuseppe Marano, Patrizia Boracchi, Elia Biganzoli, Massimo Galli.

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
