## [Decision Letter · Decision Letter 0]

24 Nov 2020

PONE-D-20-26648

Prevalence of SARS-CoV-2 in an area of unrestricted viral circulation: mass seroepidemiological screening in Castiglione d’Adda, Italy

PLOS ONE

Dear Dr. Gabriele Pagani,

Thank you for submitting your manuscript to PLOS ONE. After careful consideration, we feel that it has merit but does not fully meet PLOS ONE’s publication criteria as it currently stands. Therefore, we invite you to submit a revised version of the manuscript that addresses all  the points raised during the review process.

We look forward to receiving your revised manuscript.

Kind regards,

Daniela Flavia Hozbor

Academic Editor

PLOS ONE

Journal Requirements:

2. Thank you for including your ethics statement:  "This study was approved by University of Milan's Ethical Committee (allegato 1 Comitato Etico 21.04.20 - parere numero 35/20).".   

Please provide additional details regarding participant consent. In the ethics statement in the Methods and online submission information, please ensure that you have specified (1) whether consent was informed and (2) what type you obtained (for instance, written or verbal, and if verbal, how it was documented and witnessed). If your study included minors, state whether you obtained consent from parents or guardians. If the need for consent was waived by the ethics committee, please include this information.

"This study was realized thanks to non-conditioning financial contributions from: CISOM (Corpo Italiano di Soccorso dell'Ordine di Malta, https://www.cisom.org), FC Internazionale Milano (https://www.inter.it/), SFD s.p.a/Fondazione SAME (https://fondazionesame.it/), Emporio Armani Olimpia Milano (http://www.olimpiamilano.com/) through donations to the Dipartimento di Scienze Biomediche e Cliniche (DIBIC) of the University of Milan and to Banca Mediolanum (https://www.bancamediolanum.it/) through donation to ASST Fatebenefratelli-Sacco, Milano.

Mylan Italia s.p.a. donated the rapid tests.

The donations were used to cover the expenses related to personal protective equipments, materials, laboratory processing and personnel costs.

We note that you received funding from a commercial source: , FC Internazionale Milano, Emporio Armani Olimpia Milano

We note that one or more of the authors are employed by a commercial company: Medispa s.r.l., Synlab Italia s.p.a..

4.1. Please provide an amended Funding Statement declaring this commercial affiliation, as well as a statement regarding the Role of Funders in your study. If the funding organization did not play a role in the study design, data collection and analysis, decision to publish, or preparation of the manuscript and only provided financial support in the form of authors' salaries and/or research materials, please review your statements relating to the author contributions, and ensure you have specifically and accurately indicated the role(s) that these authors had in your study. You can update author roles in the Author Contributions section of the online submission form.

4.2. Please also provide an updated Competing Interests Statement declaring this commercial affiliation along with any other relevant declarations relating to employment, consultancy, patents, products in development, or marketed products, etc.  

Reviewers' comments:

Reviewer's Responses to Questions

**Comments to the Author**

1. Is the manuscript technically sound, and do the data support the conclusions?

Reviewer #1: Partly

2. Has the statistical analysis been performed appropriately and rigorously? 

Reviewer #1: No

3. Have the authors made all data underlying the findings in their manuscript fully available?

Reviewer #1: No

4. Is the manuscript presented in an intelligible fashion and written in standard English?

Reviewer #1: Yes

5. Review Comments to the Author

Reviewer #1: Minor issues

Write up is pretty easy read and understandable, however, please note some Syntax issues

For examples:

Vocabulary _ A different word could be used instead of diffusion that might make more sense in this context

Long Sentences_ 3rd paragraph; “Recently, lateral…..

Spelling _ “seferely” should be severely.

Technical:

Please explain why 2 different tests to run NPSs and note the differences between the two, including their sensitivity/specificity ratings.

Missing several necessary elements, including information on sample storage (temperature, location, etc.), average time intervals between sample collection and processing, and sample destruction procedures after processing was complete. Were samples given a unique identifier for blinding the study and perhaps privacy?

The reason for not measure the BMI of people under the age of 19 – This may got to Discussion.

Results:

I had to dig through the paper to find out how many people were tested for the different tests/kits. For example, on page 10, last paragraph, the authors state that 1330 people were tested using the NPS method. On page 11, under Diagnostic accuracy, the authors note that 562 subjects were selected for the CLIA serological tests and PCR.

Also, please explain well if one person might have been randomly selected for more than one test. This should be addressed in the methods section as well.

Tables design- Gender (Female) issue is confusing; how about the male data?

“The strong association observed between” should include statistical data for clarity.

6. PLOS authors have the option to publish the peer review history of their article (what does this mean?). If published, this will include your full peer review and any attached files.

Reviewer #1: No

---

## [Author Response · Author response to Decision Letter 0]

18 Dec 2020

2. Thank you for including your ethics statement: "This study was approved by University of Milan's Ethical Committee (allegato 1 Comitato Etico 21.04.20 - parere numero 35/20).". 

Please provide additional details regarding participant consent. In the ethics statement in the Methods and online submission information, please ensure that you have specified (1) whether consent was informed and (2) what type you obtained (for instance, written or verbal, and if verbal, how it was documented and witnessed). If your study included minors, state whether you obtained consent from parents or guardians. If the need for consent was waived by the ethics committee, please include this information.

Ethics statement has been amended as per journal’s requirements and moved to the “methods” section.

"This study was realized thanks to non-conditioning financial contributions from: CISOM (Corpo Italiano di Soccorso dell'Ordine di Malta, https://www.cisom.org), FC Internazionale Milano (https://www.inter.it/), SFD s.p.a/Fondazione SAME (https://fondazionesame.it/), Emporio Armani Olimpia Milano (http://www.olimpiamilano.com/) through donations to the Dipartimento di Scienze Biomediche e Cliniche (DIBIC) of the University of Milan and to Banca Mediolanum (https://www.bancamediolanum.it/) through donation to ASST Fatebenefratelli-Sacco, Milano.

Mylan Italia s.p.a. donated the rapid tests.

The donations were used to cover the expenses related to personal protective equipments, materials, laboratory processing and personnel costs.

We note that you received funding from a commercial source: , FC Internazionale Milano, Emporio Armani Olimpia Milano

We note that one or more of the authors are employed by a commercial company: Medispa s.r.l., Synlab Italia s.p.a..

4.1. Please provide an amended Funding Statement declaring this commercial affiliation, as well as a statement regarding the Role of Funders in your study. If the funding organization did not play a role in the study design, data collection and analysis, decision to publish, or preparation of the manuscript and only provided financial support in the form of authors' salaries and/or research materials, please review your statements relating to the author contributions, and ensure you have specifically and accurately indicated the role(s) that these authors had in your study. You can update author roles in the Author Contributions section of the online submission form.

4.2. Please also provide an updated Competing Interests Statement declaring this commercial affiliation along with any other relevant declarations relating to employment, consultancy, patents, products in development, or marketed products, etc. 

Funding Statement has been amended as per journal’s requirements. The corrected version is reported at the end of this letter.

Ethics statement has been moved to the right section.

 Supporting Tables have been amended

Comments to the Author

Reviewer #1: Minor issues

Write up is pretty easy read and understandable, however, please note some Syntax issues

For examples:

Vocabulary _ A different word could be used instead of diffusion that might make more sense in this context

Long Sentences_ 3rd paragraph; “Recently, lateral…..

Spelling _ “seferely” should be severely.

The authors thank you for your suggestions, syntax errors pointed out by reviewer were corrected.

Technical:

Please explain why 2 different tests to run NPSs and note the differences between the two, including their sensitivity/specificity ratings.

Due to the dramatic escalation of the epidemic in Northern Italy, not all PCR reagents were readily available. We added this explanation to the methods section and included technical specifications of both tests in the references.

Missing several necessary elements, including information on sample storage (temperature, location, etc.), average time intervals between sample collection and processing, and sample destruction procedures after processing was complete. 

Altough many of the required information were already stated in the protocol (published on a pre-print server and referenced in the methods section) we added them to the text for clarity.

Were samples given a unique identifier for blinding the study and perhaps privacy?

Samples were identified by a standard label (which included a barcode, name, surname and date of birth). Authors did not consider necessary to blind samples.

Moreover, a nominal medical report of NPSs’ results was required by Italian law, as it was necessary to file a communicable disease’s report to the health authorities.

The reason for not measure the BMI of people under the age of 19 – This may got to Discussion.

As per WHO directions (referenced in text), BMI calculation is applicable only for subjects over 19 years old (while the body mass for subjects younger than 19 should be standardized using centiles).

Results:

I had to dig through the paper to find out how many people were tested for the different tests/kits. For example, on page 10, last paragraph, the authors state that 1330 people were tested using the NPS method. On page 11, under Diagnostic accuracy, the authors note that 562 subjects were selected for the CLIA serological tests and PCR.

Also, please explain well if one person might have been randomly selected for more than one test. This should be addressed in the methods section as well.

As explained in the methods section, 562 subjects were randomized to be tested by NPS and serologies on venous blood regardless of the results of the RICT, while everyone with a positive RICT would be tested with both serologies and NPS (excluding those with a documented history of a positive NPS followed by two negatives, as per Italian laws at the time). Consequently, 1330 represents the total number of NPS, including those tested because they were part of the random sample and those with a positive RICT. It is important to note that the two groups may overlap, as a random subject could test positive to RICT.

This has been added to the text in the methods section, for clarity.

Tables design- Gender (Female) issue is confusing; how about the male data?

Tables have been amended as per reviewer request.

“The strong association observed between” should include statistical data for clarity.

Adjusted ORs were included in the text.

---

## [Decision Letter · Decision Letter 1]

21 Jan 2021

Prevalence of SARS-CoV-2 in an area of unrestricted viral circulation: mass seroepidemiological screening in Castiglione d’Adda, Italy

PONE-D-20-26648R1

Dear Dr. Gabriele Pagani,

We’re pleased to inform you that your manuscript has been judged scientifically suitable for publication and will be formally accepted for publication once it meets all outstanding technical requirements.

Kind regards,

Daniela Flavia Hozbor

Academic Editor

PLOS ONE

Additional Editor Comments (optional):

Reviewers' comments:

Reviewer's Responses to Questions

**Comments to the Author**

1. If the authors have adequately addressed your comments raised in a previous round of review and you feel that this manuscript is now acceptable for publication, you may indicate that here to bypass the “Comments to the Author” section, enter your conflict of interest statement in the “Confidential to Editor” section, and submit your "Accept" recommendation.

Reviewer #1: All comments have been addressed

2. Is the manuscript technically sound, and do the data support the conclusions?

Reviewer #1: Yes

3. Has the statistical analysis been performed appropriately and rigorously? 

Reviewer #1: Yes

4. Have the authors made all data underlying the findings in their manuscript fully available?

Reviewer #1: Yes

5. Is the manuscript presented in an intelligible fashion and written in standard English?

Reviewer #1: Yes

6. Review Comments to the Author

Reviewer #1: (No Response)

7. PLOS authors have the option to publish the peer review history of their article (what does this mean?). If published, this will include your full peer review and any attached files.

Reviewer #1: No

---

## [Editor Report · Acceptance letter]

3 Feb 2021

PONE-D-20-26648R1 

Prevalence of SARS-CoV-2 in an area of unrestricted viral circulation: mass seroepidemiological screening in Castiglione d’Adda, Italy 

Dear Dr. Pagani:

I'm pleased to inform you that your manuscript has been deemed suitable for publication in PLOS ONE. Congratulations! Your manuscript is now with our production department. 

Kind regards, 

on behalf of

Dr. Daniela Flavia Hozbor 

Academic Editor

PLOS ONE